# The Value of Micro-CT in the Diagnosis of Lung Carcinoma: A Radio-Histopathological Perspective

**DOI:** 10.3390/diagnostics13203262

**Published:** 2023-10-20

**Authors:** Serpil Dizbay Sak, Selim Sevim, Arda Buyuksungur, Ayten Kayı Cangır, Kaan Orhan

**Affiliations:** 1Department of Pathology, Faculty of Medicine, Ankara University, Ankara 06230, Turkey; 2Department of Basic Medical Sciences, Faculty of Dentistry, Ankara University, Ankara 06560, Turkey; 3Department of Thoracic Surgery Ankara, Faculty of Medicine, Ankara University, Ankara 06230, Turkey; 4Department of Dentomaxillofacial Radiology, Faculty of Dentistry, Ankara University, Ankara 06560, Turkey

**Keywords:** micro-computed tomography, pulmonary carcinoma, histopathology, light microscopy, 3D imaging, frozen section, resection margin, surgical pathology

## Abstract

Micro-computed tomography (micro-CT) is a relatively new imaging modality and the three-dimensional (3D) images obtained via micro-CT allow researchers to collect both quantitative and qualitative information on various types of samples. Micro-CT could potentially be used to examine human diseases and several studies have been published on this topic in the last decade. In this study, the potential uses of micro-CT in understanding and evaluating lung carcinoma and the relevant studies conducted on lung and other tumors are summarized. Currently, the resolution of benchtop laboratory micro-CT units has not reached the levels that can be obtained with light microscopy, and it is not possible to detect the histopathological features (e.g., tumor type, adenocarcinoma pattern, spread through air spaces) required for lung cancer management. However, its ability to provide 3D images in any plane of section, without disturbing the integrity of the specimen, suggests that it can be used as an auxiliary technique, especially in surgical margin examination, the evaluation of tumor invasion in the entire specimen, and calculation of primary and metastatic tumor volume. Along with future developments in micro-CT technology, it can be expected that the image resolution will gradually improve, the examination time will decrease, and the relevant software will be more user friendly. As a result of these developments, micro-CT may enter pathology laboratories as an auxiliary method in the pathological evaluation of lung tumors. However, the safety, performance, and cost effectiveness of micro-CT in the areas of possible clinical application should be investigated. If micro-CT passes all these tests, it may lead to the convergence of radiology and pathology applications performed independently in separate units today, and the birth of a new type of diagnostician who has equal knowledge of the histological and radiological features of tumors.

## 1. Introduction

Histopathological imaging and its limitations

The routine examination method for surgical specimens resected for pulmonary carcinoma is histopathological examination, as in all other human tumors. This examination provides information on tumor type; tumor size; lymphatic, blood vessel, and perineural invasion; extent of the invasion to the adjacent tissues and organs; and lymph node (LN) metastasis. In this method, specimens are examined macroscopically by a histopathologist according to established protocols and the samples taken for microscopic examination are routinely fixed in 10% formalin for 6–48 h. Following tissue processing steps, the samples are embedded in paraffin. Thin (4–6 μm) sections cut from the blocks are placed on glass slides and are routinely stained with hematoxylin-eosin (HE) and/or other stains and evaluated using a light microscope. Formalin-fixed and paraffin-embedded (FFPE) tissue blocks can be stored for many years and other (e.g., immunohistochemistry, in situ hybridization, polymerase chain reaction-based) studies can be performed, as required. The information given in the pathology report constitutes the mainstay of cancer management. Histopathology is also used to direct surgery in the intraoperative period. In this examination, the tissue is quickly frozen and sectioned while the patient is under anesthesia. By examining the frozen sections, a pathologist can give information on the tumor type, surgical margin status and/or infiltrated tissues/organs, within an average time of 15–30 min, and the operation can be planned accordingly [1]. For example, in the presence of a tumor positive or near parenchymal resection margin, the thoracic surgeon may decide to perform a lobectomy instead of a sub-lobar resection.

Although histopathology is a time-honored method and constitutes the gold standard for the examination of human tumors, both conventional glass slides and their relatively new alternative, digital whole slide images (WSIs) obtained by scanning the glass slides, provide only two-dimensional (2D) information on a tumor, which is in fact a three-dimensional (3D) entity [2,3]. Although 3D images can be reconstructed from serial WSIs, this is difficult and time-consuming, and it requires the whole paraffin block to be consumed, thereby disrupting the integrity of the tissue examined [4,5,6]. Additionally, in a histopathological examination, the amount of submitted tissue and the size of the tissue cassettes limit the material that is examined. Furthermore, bony and calcified tissues must be decalcified, and staples must be removed for sectioning to prepare the glass slides for microscopic examination.

In this article, it was questioned whether micro-computed tomography (micro-CT), which is a 3D technique, could replace the traditionally used histopathological examination to provide the necessary information for the management of pulmonary carcinoma. For this purpose, the results obtained in studies conducted with micro-CT in clinical specimens from pulmonary and other human tumor types were investigated, and the possible role of this new tool in the evaluation of lung cancer specimens was examined.

## 2. Micro-CT as an Emerging Imaging Modality in Human Tumor Pathology

Micro-CT is a relatively new tomographic technique with a voxel size volumetrically almost one million times smaller than that of conventional computed tomography (CT) [7,8], which utilizes geometrically cone-shaped beams for reconstruction and back-projection processes. The 3D images that are obtained with high-resolution micro-CT technology allow researchers to collect both quantitative and qualitative information on various types of samples in many fields [9,10,11,12,13]. Micro-CT is generally used to obtain precise information about the internal structure of materials. It is known to be used in the industrial field and in materials sciences such as engineering, microprocessor production, dealing with materials such as stone and metal [14,15,16], and also in the examination of human tissues [17,18,19,20,21]. Although micro-CT is a method that is quite different from histopathologic procedures, as pathologists currently know it, this ever-evolving modality is very interesting for the surgical pathologist as it provides a much higher resolution than conventional radiology and has the potential to find answers to some questions that are the subject of surgical pathology examinations. The tissue detail approaching to that of the light microscope, thanks to the high resolution of micro-CT; the possibility of 3D examination, and the fact that the tissue remains intact at the end of the examination, suggest that this method may have a place in the surgical pathology laboratories of the future. 

Although micro-CT is far from being a method that is routinely used in the diagnosis of human diseases, 636 articles were present in a query made on PubMed^®^ with the keywords (“micro-CT” OR “X-ray micro-tomography” OR “micro-computed tomography”) AND (“lung*” OR “pulmonary”) on the 2 September 2023. According to the PRISMA Extension for Scoping Reviews (PRISMA-ScR) [22], 27 reviews and 7 case reports/letters were removed. Non-English (*n* = 10) records and studies with the species option marked as other than “human” in the database (*n* = 383) were excluded. Of the 209 reports assessed for eligibility, only three studies were performed on human lung carcinoma specimens; the remaining 206 were non-tumoral, animal studies, or irrelevant. Apart from these three studies, most of the other studies represent the research results on tumoral and non-tumoral lung pathologies performed on laboratory animals, as this promising novel imaging modality may be useful in understanding various lung diseases. In a recent scoping review by Bompoti et al., 37 studies on the clinical applications of micro-CT for the tissue-based diagnosis of lung diseases, conducted between 2005 and 2021, were evaluated [23]. In this study, the authors concluded that micro-CT-based volumetric measurements and qualitative evaluations of pulmonary tissue could be used for the clinical management of various lung diseases. Considering the possible contributions of this method to the understanding and diagnosis of lung tumors, studies focusing on lung tumors performed on clinical material are surprisingly few. The main features of the three studies conducted on clinical specimens of lung cancer are summarized in Table 1. 

Although studies on clinical pulmonary specimens are very few, studies on other clinical tumor specimens can be a guide for future studies on lung tumors. A total of 1052 articles were found in a query made on PubMed^®^ with the keywords (“micro-CT” OR “microCT” OR “micro-computed tomography”) AND (“tumor” OR “carcinoma”) on the 2 September 2023. In accordance with PRISMA-ScR [22], 17 reviews and 8 case reports/letters were removed. Non-English (*n* = 25) records and studies with the species option marked as other than “human” in the database (*n* = 567) were excluded. Of the 435 reports assessed for eligibility, only 26 studies were performed on human non-pulmonary tumor specimens; the remaining 409 were non-tumoral, animal studies, or irrelevant. The important features of these studies, most of which were conducted on breast carcinoma, are summarized in Table 2.

## 3. Potential Contribution of This Imaging Modality to the Examination of Pulmonary Carcinoma Specimens

### 3.1. Examination Steps in Which Micro-CT Can Be Used

While there are many micro-CT studies investigating tumors in live and sacrificed animals [52,53,54,55,56,57,58,59,60,61,62,63,64,65], this technique can only be used for the examination of ex vivo specimens in the context of human tumors, since the radiation dose used in micro-CT examination is very high compared to a conventional CT [7,66,67]. Both fresh or formalin-fixed specimens can be successfully scanned with micro-CT [26,35,41,42,68,69,70,71,72] (Figure 1, Figure 2 and Figure 3; Appendix A); however, the protocols to be employed need to be optimized to obtain the best results. In a study using fresh lung specimens, it was reported that artifacts caused by insufficient air–tissue contrast or tissue deflation could sometimes obscure the target lesion, but these artifacts could be reduced by making some adjustments to the protocol [26]. In many studies on paraffin blocks, X-ray attenuating solutions that contain heavy ions such as tungsten, iodine, silver, or barium were used, due to the low image contrast between tissue and paraffin [73,74,75,76]. However, in addition to the increased pre-processing time, these procedures have the potential to hinder tissue quality and interfere with subsequent studies such as immunohistochemistry and in situ hybridization. To circumvent this problem, Katsemenis et al. proposed a soft tissue-optimized micro-CT workflow that can be validated through routine histologic techniques [77]. Although their performance is inferior to those of full size micro-CT instruments, the spread of benchtop laboratory micro-CT units with a longest dimension of about 1000 mm and weight of 250–400 kg [78], will facilitate the entrance of this new technique into pathology laboratories. Furthermore, with the development of more powerful X-ray tubes and better detectors, we can expect to have better resolutions that can be used to examine both fresh and FFPE human tissues in the near future. However, to be able to use the micro-CT efficiently, a data/image management system to capture the images and make that information available to the end user is necessary [35]. Development of deep neural networks to analyze whole tissues and blocks in a short time, as shown by Ohnishi et al. [6], will improve the accuracy and usability of micro-CT in human tumors. Artificial intelligence can substantially reduce the hands-on time required to perform lung tumor segmentation and reduce bias and error associated with manual segmentation of micro-CT images [79]. As a combined result of all these developments, it may be possible to evaluate histopathological features using this new technology both during the intraoperative evaluation of lung tumors, during the macroscopic examination and sampling of the surgical material after surgery, and also in FFPE archive material.

### 3.2. Micro-CT Cannot Replace Microscopy in Its Current Capacity

Studies with micro-CT on human lung tumors are very few. Nakamura et al. showed that micro-CT can act as a bridge between high-resolution computed tomography and microscopy in their study on 10 pulmonary adenocarcinomas with ground glass opacity [25]. According to this study, thickened alveolar walls corresponded to adenocarcinoma with lepidic growth pattern, and thin alveolar walls corresponded to normal lung [25]. Virtual histological images can be obtained via micro-CT, with the application of special ex vivo protocols [31,80,81,82,83]. There are a few studies on human tissues showing that it is also possible to obtain digitally colored images comparable to HE-stained slides from routinely prepared FFPE tissue blocks [24,34,84] (Figure 2A inset). However, Xu et al. stated that it is impossible to distinguish pseudopapillae from true papillae, lymphatic invasion from blood vessel invasion, and different cell types from each other in thyroid cancers using micro-CT [34]. Although it is very appealing to be able to examine a tumor via uncut paraffin blocks and from the desired plane of section, benchtop micro-CT has not yet reached the resolution level that can be achieved with a light microscope. Accordingly, it seems impossible, for the time being, to determine the type or pattern of lung carcinoma with virtual images obtained via micro-CT. However, in the study by Cangır et al. on paraffin blocks of lung adenocarcinoma, it was shown that the structural parameters (e.g., percent object volume, structure thickness, structure linear density, connectivity, connectivity density, intersection surface, structure model index, open porosity percentage, and closed porosity percentage) calculated using the micro-CT images were dissimilar in the tumoral and non-tumoral regions of interest (ROIs). They concluded that non-tumoral and tumoral areas in the paraffin blocks could be distinguished from each other using quantitative data obtained from micro-CT [24]. Similarly, in another study, bone marrows with multiple myeloma involvement and bone marrows without disease could be separated from each other via micro-CT structural parameters, namely bone volume fraction, bone surface to bone volume ratio, and trabecular thickness [32]. In addition, according to the results of an unpublished study conducted by our group, different types of solid patterned lung tumors may differ from each other in terms of some structural parameters [85]. However, there is no published data on the discrimination of different pulmonary tumor types or adenocarcinoma patterns using quantitative parameters obtained via micro-CT or using micro-CT images. 

### 3.3. Tumor Size and Volume Estimation via Micro-CT

As with cancers of other organs, the TNM classification is a globally recognized, anatomically based standard for classifying the extent of lung cancer [86]. Micro-CT can be useful in a more accurate determination of the T descriptor [44]. The T category describes the largest diameter of the primary tumor as well as the tumor invasion to adjacent structures. However, since tumors are not perfectly spherical and the tumor margin is irregular, especially in malignant tumors, 3D tumor volume may be a better predictor [87]. In addition, tumor volume is a parameter used to evaluate the performance of new therapies in lung cancer research [88,89,90]. In a mouse model, Bidola et al. demonstrated that a quick estimation of tumorous tissue volume could be performed using micro-CT images [91]. It has also been demonstrated that micro-CT can be used to estimate tumor volume in human breast carcinomas [43,44]. However, the success of micro-CT in this regard may not be the same in all tumor types. In Sarraj et al.’s study, there was no agreement between micro-CT and pathology in estimating and staging tumor size for invasive lobular carcinoma, in contrast to a strong agreement between the two methods for invasive ductal carcinoma [44]. Such discrepancies can be expected in some tumors where the tumor invades the parenchyma, without destroying or replacing it. In the lung, lepidic adenocarcinomas may hypothetically present more difficulty in this regard. Also, lung carcinomas developing on a previously fibrotic and destroyed lung parenchyma may be prone to errors in volume estimation via micro-CT. 

### 3.4. Detection of LN Metastases Using Micro-CT

In lung cancers undergoing surgical treatment, LN metastasis is the most important factor for determining the survival and adjuvant treatment decision. Therefore, it is critical to determine the LN metastasis accurately. It has been shown that multiphase contrast-enhanced micro-CT is feasible for the evaluation and monitoring of lung cancer metastases in small animal studies [92,93,94]. Another study indicates that micro-CT can be used as a method for diagnosing metastatic LNs with a clinical N0 status in mice [95]. Micro-CT applications in LN metastases in human tumors are scarce. One study showed that micro-CT could accurately locate metastatic foci in regional LN metastases of oral squamous cell carcinomas [31]. In another study, it was shown that micro-CT could accurately evaluate the volume of nodal metastases in thyroid cancers [34]. In a pilot study by our group, using FFPE blocks of metastatic lung carcinoma to LNs, it was shown that metastatic and non-metastatic regions could be distinguished using structural parameters via micro-CT [96]. In the current surgical pathology protocols for the examination of pulmonary carcinoma, it is necessary to report the localization (LN station), number and extra-nodal invasion status of the metastatic LNs. Considering the limited available data, micro-CT emerges as an imaging modality that could provide this information and more on resection specimens.

### 3.5. Possible Role of Micro-CT in the Intraoperative Examination of Lung Tumors

With the development of radiological imaging methods, the number of pulmonary tumors detected at an early stage has increased, as in other organs. Accordingly, the number of sub-lobar resections has expanded. In the surgical resection of early stage cancers, it is very important to ensure that the tumor is removed with a safe surgical margin. The intraoperative surgical margin examination may change the scope of surgery in cases where a sub-lobar resection is planned initially. Traditionally, the surgical margin examination is based on the gross examination of the resection material and the microscopic evaluation of a few HE-stained frozen sections, representing the nearest macroscopic resection margin. As noted by DiCorpo et al. in their article on the role of micro-CT in imaging breast cancer specimens, a breast specimen has an average surface area of 45 cm^2^, which corresponds to an average of 20 million light microscopic high-power fields [35]. Although resection margins may be smaller in most lung wedge resection and segmentectomy specimens, it is impossible to entirely examine an area of this size intraoperatively. In this context, the use of a new method that can examine the whole tissue in 3D, in an acceptable period of time, in conjunction with the traditional method, can increase the accuracy of diagnosis. In Troschel et al.’s study examining 22 benign and malignant lung lesions, it was found that the target lesions could be localized in the majority of specimens and high-resolution morphological data regarding the entire lesion could be obtained with micro-CT [26]. In their study, Troschel et al. determined the distance to the nearest surgical margin in 10 specimens and they found that the agreement of micro-CT with final pathology was good [26]. 

Spread through airspaces (STAS) is defined as tumor cells within airspaces in the lung parenchyma beyond the edge of the main tumor [97]. STAS in adenocarcinoma may be composed of micropapillary structures, solid nests of tumor cells filling alveoli, or discohesive single cells [97]. Recently, a 3D study showed that STAS tumor cells can be found attached to air spaces rather than free-floating in them [98]. There are a number of independent studies showing that STAS is a predictor of poor clinical outcome both in resected lung adenocarcinomas and other types of lung carcinomas [97,99,100,101,102]. In addition, in patients with STAS, limited resection may have a higher risk of recurrence than lobectomy [97,99]. Knowing about the presence of STAS intraoperatively in a given tumor, a thoracic surgeon may decide to perform a lobectomy instead of a sub-lobar resection. On the other hand, recognition of STAS on a frozen section is a challenging task for the histopathologist [103]. In addition to the difficulty of distinguishing STAS from artifacts, in most clinical settings, it is often impossible to accurately identify STAS within a reasonable time frame, with the limited number of sections that can be examined during an intraoperative consultation. Although currently there are no studies on this particular subject, micro-CT could hypothetically be useful in detecting STAS, as a method that can obtain 3D images within a reasonable time frame, rather than the 2D HE-stained slides that are traditionally used in intraoperative consultations. Even if the images obtained during the intraoperative consultation do not reach a sufficient resolution level to give a definite diagnosis of STAS, micro-CT may enable a more accurate determination of suspicious areas that require further histological examination.

In all examinations performed intraoperatively on the tissues of a patient under anesthesia, the procedure time is a critical factor. The result should be given as soon as possible within a clinically relevant time period and the type and extent of the surgery should be planned accordingly. In the study by Troschel et al., the average time for image acquisition was 13 min, and in the study by Dicorpo et al. it was stated as 8–10 min [26,35]. In another study on breast specimens, the combined acquisition and reconstruction time was reported as under 4 min [41]. The type of scanner, scanning conditions, and the voxel size can cause variations in the investigation time. When such a method is routinely used, image acquisition and reconstruction times can be shortened by obtaining fewer images [26]. Additionally, in the near future, it may be possible, with the help of new technical developments, to reduce this period further, while preserving or even improving the image quality.

## 4. Impact of Micro-CT on Conventional Histological Examination

The study by Troschel et al. on 22 benign and malignant lung lesions showed that the specimens examined via micro-CT were adequate for further histopathological analyses [26]. In this study, no difference was found between the group examined via micro-CT and the control group in terms of histological quality. In this study, it was also shown that the micro-CT examination did not affect molecular analyses [26]. Although it is encouraging to know that this new method does not seem to have a negative effect on molecular analyses, more data is required on this particular subject, since molecular investigations are required in a good number of lung carcinomas for the detection of present and evolving therapeutic targets (e.g., EGFR mutations, ALK and ROS1 translocations, and many others) and the high-dose radiation administered for micro-CT examination may potentially have deteriorating effects on the genetic material of the tumor cells.

## 5. What Is on the Horizon for Pulmonary Diagnosticians Regarding Micro-CT?

At present, the resolution of micro-CT has not yet reached the levels that can be obtained with a light microscope, although it is much improved compared to conventional radiological methods. Therefore, it is not yet possible to replace the histopathological examination required in lung cancer diagnosis and management with micro-CT. However, when we consider the high speed of technological developments in our era, and that the wavelength of X-rays is 0.01 to 10 nanometers, we can appreciate that it is only a matter of time before micro-CT devices reaching and exceeding the resolution capacity of light microscopes used in pathology laboratories become available [35]. In addition, the ability of micro-CT to provide a 3D image in any plane of section without disturbing the integrity of the specimen suggests that it has a potential to be used as an auxiliary technique in intraoperative consultations and routine histopathological examinations of pulmonary tumor specimens. It can be predicted that in lung resection materials, micro-CT can facilitate the pathologist’s work, especially in surgical margin examination, the evaluation of tumor invasion in the entire specimen, and the calculation of primary and metastatic tumor volume. With the future developments in micro-CT technology, it can be expected that the image resolution will gradually improve, and the examination time will decrease. We can also hope that the future image analysis software accompanying micro-CT equipment will be more user-friendly to meet the requirements of non-engineers working in the field. Another factor that will limit the use of micro-CT in pathology laboratories is the cost. Micro-CT scanner systems are widely used in different areas. Micro-CT scanners are composed of three main components; the X-ray source, the detector, and the rotary stage [104]. The cost of the system is dependent mainly on the field of view and the energy of the X-ray tube, and the systems can cost from USD 100,000 to over USD 1,000,000 depending on the components used [105]. The use of a medium-sized high-definition detector reduces the overall cost of the system and thus this technique could be suitable for routine use [105]. 

On the other hand, although the possible contributions of micro-CT to pulmonary tumor diagnosis are very tempting, the data are by no means sufficient to replace the current and well-established gold standard histopathological diagnostic method. It is very important to investigate the safety, performance, and cost effectiveness of this imaging modality rigorously in the areas of possible clinical application with research teams including pathologists, radiologists, pulmonologists, and thoracic surgeons.

## 6. Conclusions

Considering the developments in molecular and digital pathology in the last few decades, it is very probable that micro-CT will emerge as a new area of challenge and development both for pulmonary pathologists and radiologists. As we have always done before, we can embrace this new modality and incorporate it into our daily practices, if we are convinced that it will be beneficial to our patients.

## 7. Future Directions

Pathology and radiology are two separate disciplines that complement each other in the diagnosis and treatment of lung tumors. Both examination methods have their inherent limitations. Although histopathology is a very high-resolution method, it is prone to sampling error, can evaluate a small and selected region, and produces only 2D images. Conventional CT, on the other hand, offers the opportunity to examine the entire lung in 3D in a lung cancer patient. However, it has a much lower resolution, despite the development of newer methods such as high-resolution CT, which considerably increases the image resolution. Surgical pathologists and radiologists who have mastered the microscopic and radiological appearance of diseases, respectively, and traditionally work independently of each other, may need to work together in the near future. The surgical resection specimens performed for a tumor can be examined in three dimensions with micro-CT in the intraoperative and postoperative period, and prognostic parameters such as tumor size, tumor volume, tumor spread, the status of the surgical margins, and the presence of metastasis can be evaluated in detail, even before gross examination and sampling. Thus, the sampling of surgical specimens for further microscopic examination can be performed more efficiently. Determining the variations of the structural parameters of different lesion and tumor types can even lead to tumor typing with micro-CT using artificial intelligence further down the line. Although it does not align with our current practices, in the not-too-distant future, “radio-histopathologists” may also be involved in the examination of lung tumors. 

## Figures and Tables

**Figure 1 diagnostics-13-03262-f001:**
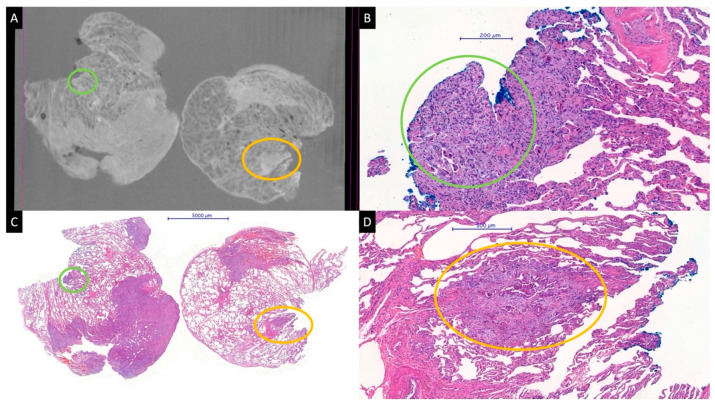
Micro-CT image from a paraffin block (**A**) and whole slide HE images (**B**–**D**) of a pulmonary adenocarcinoma. A satellite tumor nodule of about 1000 μm, very near to the blue-stained parenchymal resection margin, is perfectly visible in the micro-CT image (yellow circles). A smaller tumor nodule (green circles) at the margin is barely visible on the micro-CT image (**A**). The specimens were scanned with a high-resolution desktop micro-CT system (Bruker micro-CT Systems 1275, Kontich, Belgium). The scanning parameters used were 35 kVp, 231-mA, without filter, 16 µm pixel size, rotation at 0.2 steps with 360°. The mean time of scanning was around 30 min. NRecon (ver. 1.7.4, Bruker micro-CT Systems, Kontich, Belgium) software was used for the reconstruction. CTAn (v. 1.20.3 Bruker micro-CT Systems, Kontich, Belgium) software was used for the 3D image analysis. CTVox (v. 3.3.0 Bruker micro-CT Systems, Kontich, Belgium) was used for visualization of the samples for images.

**Figure 2 diagnostics-13-03262-f002:**
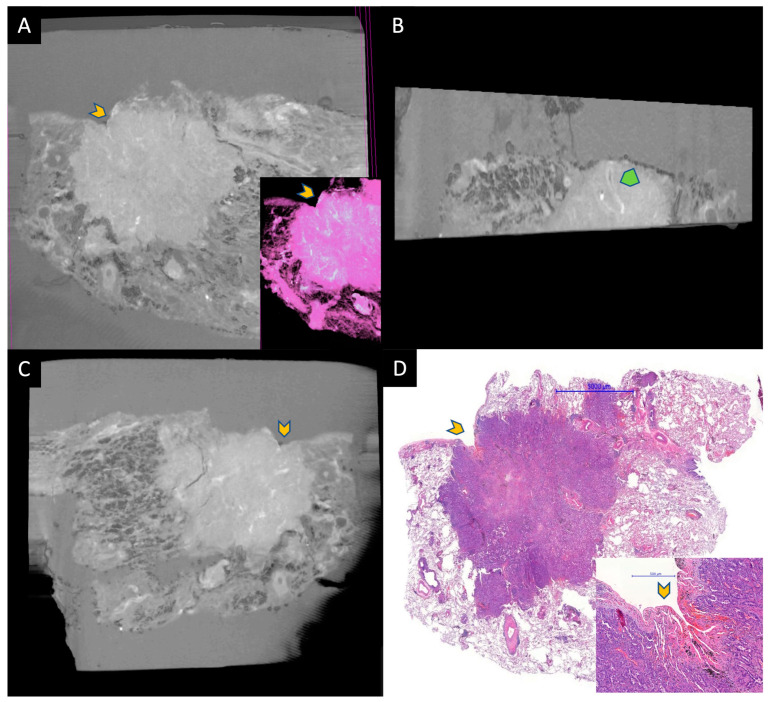
A small adenocarcinoma located under the visceral pleura. (**A**–**C**) are micro-CT images taken from the paraffin block, and (**D**) is HE-stained WSI for comparison. Gray area around the tissue in micro-CT images represents paraffin. Yellow chevrons represent the retracted pleura by the tumor. A is an image taken from the surface of the paraffin block and the inset is a pseudo-HE. B is taken at a 90 angle to the surface. Tumor and normal lung can be discerned easily. The green chevron represents an artery within the tumor. Image C shows a deeper aspect of the tumor parallel to the surface. The specimens were scanned with a high-resolution desktop micro-CT system (Bruker micro-CT Systems 1275, Kontich, Belgium). The scanning parameters used were 35 kVp, 231-mA, without filter, 16 µm pixel size, rotation at 0.2 steps with 360°. The mean time of scanning was around 30 min. NRecon (ver. 1.7.4, Bruker micro-CT Systems, Kontich, Belgium) software was used for the reconstruction. CTAn (v. 1.20.3 Bruker micro-CT Systems, Kontich, Belgium) software was used for the 3D image analysis. CTVox (v. 3.3.0 Bruker micro-CT Systems, Kontich, Belgium) was used for visualization of the samples for images.

**Figure 3 diagnostics-13-03262-f003:**
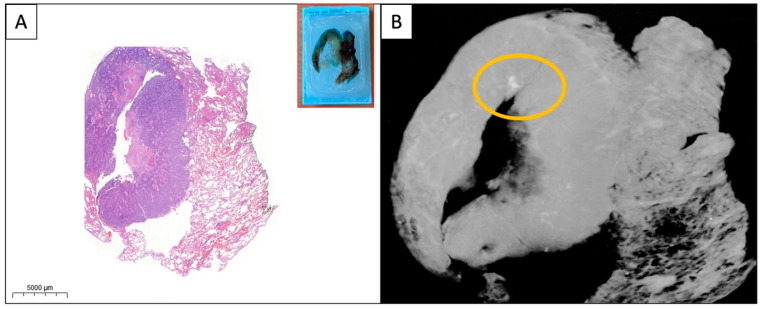
(**A**,**B**) WSI and micro-CT image of the paraffin block from a solid pulmonary adenocarcinoma. A small area of calcification (yellow circle) is visible on the micro-CT image which is absent on the HE-stained WSI. The specimens were scanned with a high-resolution desktop micro-CT system (Bruker micro-CT Systems 1275, Kontich, Belgium). The scanning parameters used were 35 kVp, 231-mA, without filter, 16 µm pixel size, rotation at 0.2 steps with 360°. The mean time of scanning was around 30 min. NRecon (ver. 1.7.4, Bruker micro-CT Systems, Kontich, Belgium) software was used for the reconstruction. CTAn (v. 1.20.3 Bruker micro-CT Systems, Kontich, Belgium) software was used for the 3D image analysis. CTVox (v. 3.3.0 Bruker micro-CT Systems, Kontich, Belgium) was used for visualization of the samples for images.

**Table 1 diagnostics-13-03262-t001:** Studies using micro-CT on clinical specimens of lung carcinoma.

Main Author, Year, Country	Number of Specimens and Material Type	Micro-CT Scanner	Study Design and Results	Main Outcome
Kayi Cangir, A. [24] 2021, Turkey	3 FFPE block	Skyscan 1275 (Bruker, Kontich, Belgium)	Twenty one regions of interest (ROIs) from three adenocarcinomas (two cases of predominantly acinar and one case of predominantly micropapillary pattern) and surrounding pulmonary parenchyma were compared regarding various structural parameters calculated from micro-CT images. All parameters (percent object volume, structure thickness, structure linear density, connectivity, connectivity density, intersection surface, structure model index, open and closed porosity) were significantly different regarding the tumoral and non-tumoral ROIs.	Tumoral and non-tumoral areas can be separated from each other by means of structural parameters calculated via micro-CT.
Nakamura, S. [25] 2020, Japan	10 Fixed tissue	InspeXio SMX-100CT (Shimadzu, Kyoto, Japan)	Ten resected human lungs with primary adenocarcinoma (adenocarcinoma in situ, minimally invasive, lepidic-predominant and papillary predominant) were scanned after fixation. The thickness of the alveolar walls of the normal lung and the cancer area of which alveoli might be replaced by tumor cells, was measured and compared with histopathological images. The area of thickened alveolar walls on the micro-CT corresponded well with the lepidic growth patterns of adenocarcinoma. Micro-CT images could be correctly divided by (the thickness of) alveolar walls into normal lung and adenocarcinoma.	Pulmonary adenocarcinomas with lepidic pattern can be differentiated from normal lung parenchyma via micro-CT-based measurements. Further detailed investigations are needed to make comparable histological diagnoses using micro-CT.
Troschel, F. [26] 2019, USA	22 Fresh tissue	1. Skyscan 1275 (Bruker, Kontich, Belgium) 2. XT H 225 (Nikon Metrology Inc., Brighton, MI, USA)	Twenty-two lung specimens with a presumptive diagnosis of lung cancer from 21 patients were scanned. Images were assessed to determine image quality, lesion size, and distance to the nearest surgical margin. Micro-CT measurements were compared to histopathologic measurements. Scanned specimens were indistinguishable from a control group of non-imaged lung specimens in terms of image quality. The agreement of micro-CT with the final pathology was good concerning the lesion size and the distance to surgical margin.	Micro-CT-scanned specimens were adequate for subsequent histopathologic analysis including molecular assays. Micro-CT can be an auxiliary method in the analysis of surgical margins in the fresh lung specimens with tumors.

**Table 2 diagnostics-13-03262-t002:** Studies using micro-CT in clinical tumor specimens on various tumor types.

Main Author, Year, Country	Number and Type of Specimens	Micro-CT Scanner	Main Outcome
Streeter, S.S. [27] 2023, USA	100 Fresh tissue Breast carcinoma	PerkinElmer (Hopkington, MA, USA)	Micro-CT scanning alone is ineffective for margin assessment in complex cases. Adjuvant methods of margin assessments combined with micro-CT may be beneficial.
Sakamoto, H. [28] 2022, USA	9 Fresh and fixed Gastrointestinal system tumors	Nikon XT H 160 MedX Alpha, Nikon Metrology NV (Leuven, Belgium)	Micro-CT could delineate the extent of the lesion in endoscopic submucosal dissection specimens.
Brahimetaj, R. [29] 2022, Belgium	94 (Fresh?) Vacuum-assisted stereotactic biopsies with microcalcifications Breast lesions	Skyscan 1076 (Bruker, Kontich, Belgium)	Texture features of breast microcalcifications that are obtained via micro-CT have high discriminating power to classify individual microcalcifications as benign or malignant in breast lesions.
Alkalay, R.N. [30] 2021, USA	45 Fresh tissue Vertebra biopsies from metastatic tumors	Scanco Medical (Brüttisellen, Switzerland)	Micro-CT may determine bone mineral density, strength, and stiffness better than qualitative clinical classification of bone lesions.
Xia, C.W. [31] 2021, China	4 Fixed tissue Lymph nodes with suspected metastasis	Hiscan XM (Suzhou Heisfeld Information Technology Co., Ltd., Suzhou, China)	Micro-CT can accurately detect the 3D location of metastatic foci in regional lymph nodes in oral squamous cell carcinomas. This method will overcome the blind sampling deficiency in traditional pathological examination and thus reduce the rate of overlooked metastatic lymph nodes.
Chen, L. [32] 2021, China	31 Fixed tissue Bone marrow tru-cuts	Hiscan XM (Suzhou Heisfeld Information Technology Co., Ltd., Suzhou, China)	Bone marrows with multiple myeloma involvement and bone marrows without disease can be separated from each other via micro-CT structural parameters.
Ohnishi, T. [6] 2021, USA	3 Colon carcinoma Fresh whole tissue and FFPE blocks	Nikon Metrology NV (Hertfordshire, UK)	A deep neural network was developed to process micro-CT images, which was utilized successfully to segment vessels.
Xia, C.W. [33] 2020, China	21 Fixed tissue Tongue	Hiscan XM (Suzhou Heisfeld Information Technology Co., Ltd., Suzhou, China)	Micro-CT with improved image quality can accurately position the tumor, show the surgical margin in squamous cell carcinomas of the tongue, and therefore can be used intraoperatively.
Xu, B. [34] 2020, USA	28 FFPE block Thyroid tumors	Nikon Metrology NV (Leuven, Belgium)	3D imaging can reveal the tumor burden and tumor features that may change the tumor diagnosis, such as capsule invasion in thyroid tumors.
DiCorpo, D. [35] 2020, USA	173 Whole fresh tissue in container and “bread-loaffed” slices in cassettes Breast	1. SkyScan1173, (Bruker, Kontich, Belgium) 2. SkyScan1275 (Bruker, Kontich, Belgium) 3. XT H 225, Nikon Metrology Inc. (Brighton, MI, USA)	Micro-CT can be more precise than pathologists in detecting surgical margin positivity.
Ferstl, S. [36] 2019, Germany	2 Fixed tissue Kidney tumors	IQon Spectral CT, Philips Healthcare	The relationship with the capsule, Bowman’s space, and vascular structures in kidney tumors can be revealed more clearly and therefore this technology can be used in the routine pathology.
Giuliani, A. [37] 2019, Italy	15 Fixed tissue from normal myometrium and uterine leiomyoma from 2 patients	SYRMEP, Elettra Synchrotron Facility (Basovizza, TS, Italy)	The quantitative morphometric analysis via micro-CT has a potential to investigate and understand soft tissue lesions.
Janssen, N.N.Y. [38] 2019, Netherlands	100 Fresh tissue Breast	Skyscan 1275 (Bruker, Kontich, Belgium)	Micro-CT analysis of breast excision specimens for examination of resection margins showed moderate accuracy and considerable interobserver variation. It could have a place in clinical use, if the positive predictive value and sensitivity are improved.
Baran, P. [39] 2018, Australia	10 Benign and malignant cores taken from FFPE blocks of two mastectomy specimens Breast	SYRMEP, Elettra Snychrotron (Trieste, Italy)	Micro-CT can provide high-resolution images, at a near-histological level. Detailed architectural assessment of tissue may increase sensitivity and specificity in breast cancer diagnosis, as compared to current imaging practices.
Rabelo, G.D. [40] 2018, Brazil	20 Fixed tissue Mandibulectomy specimens for tumor resection	SkyScan 1174 (Bruker, Kontich, Belgium)	The cortical bone microarchitecture changes in the proximity of the squamous cell carcinoma.
McClatchy, D.M. [41] 2018, USA	32 Fresh tissue Breast malignant and benign lesions	IVIS SpectrumCT, PN 128,201, (PerkinElmer, Hopkington, MA, USA)	Micro-CT can be a method with rapid results in the surgical margin analysis of breast lesions.
Qiu, S.Q. [42] 2018, Netherlands	30 Fresh tissue Breast in situ and invasive carcinoma	SkyScan 1275 (Bruker, Kontich, Belgium)	Micro-CT scanning is a promising technique for intraoperative margin assessment in breast carcinoma and could potentially reduce the number of reoperations.
Tang, R. [43] 2016, USA	50 Fresh tissue Breast	Skyscan 1173 (Bruker, Kontich, Belgium)	Intraoperative micro-CT can be used to evaluate whether the entire tumor is within the lumpectomy specimen by comparing tumor size on micro-CT with preoperative images.
Sarraj, W.M. [44] 2015, USA	72 Fresh tissue Breast carcinoma	Skyscan 1173 (Bruker, Kontich, Belgium)	Measurements with micro-CT for T stage in invasive ductal carcinomas were highly correlated with pathological measurements.
Willekens, I. [45] 2014, Belgium	11 Fixed tissue Breast tru-cuts	Skyscan 1076 (Bruker, Kontich, Belgium)	Micro-CT shows promise as a valuable tool for understanding the morphological characteristics of benign and malign microcalcifications.
Jensen, T.H. [46] 2013, Denmark	17 FFPE block Lymph nodes from breast carcinoma cases	ID19, European Synchrotron Radiation Facility (ESRF) (Grenoble, France)	Micro-CT can accurately detect density variations to obtain information regarding axillary lymph node involvement status.
Tang, R. [47] 2013, USA	103 Fresh tissue Breast tissue and lymph node	Skyscan 1173, (Bruker, Kontich, Belgium)	3D examination provided information about the localization of masses and calcifications relative to margins in intact lumpectomy specimens and it is a potentially useful tool allowing real-time analysis of tumor location in breast specimens and surgical margins. It may also be useful for assessing sentinel lymph nodes and mastectomy specimens.
Tang, R. [48] 2013, USA	25 Fresh tissue Breast	Skyscan 1173, (Bruker, Kontich, Belgium)	Micro-CT is a promising method for intraoperative assessment of margins in breast carcinoma.
Gufler, H. [49] 2011, Germany	15 Fixed tru-cut biopsies Breast tru-cuts (10 in situ/invasive breast carcinomas, 5 benign breast tissue)	SkyScan 1072 (Aartselaar, Belgium)	Micro-CT images are comparable to low-power HE images and the differentiation of breast tissue components via micro-CT is possible.
Chappard, D. [50] 2010, France	247 Fixed tissue Transiliac bone biopsies from suspected malignancies	Skyscan 1072 (Kontich, Belgium)	Micro-CT can be used in the immediate post-biopsy period to recognize malignancy in a shorter period of time than conventional pathology. However, it should be supported with histopathological analysis.
Langheinrich, A.C. [51] 2008, Germany	12 Fresh tissue Bone specimens from primary bone tumors	Skyscan 1072 (Kontich, Belgium)	Micro-CT is capable of differentiating osteosarcoma and chondrosarcoma.

## Data Availability

Data sharing not applicable.

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
