# Peer review of "The Value of Micro-CT in the Diagnosis of Lung Carcinoma: A Radio-Histopathological Perspective"

_diagnostics, 2023, doi:10.3390/diagnostics13203262_

Round 1

Reviewer 1 Report

The following comments are,

Introduction section doesn’t cover the basics of CT and applications. How the readers will understand about this study?

What is the aim of this study? Are you delivering any outcome through this study?

Many innovative techniques introduced through CT approach. Those are not covered in this study.

The statistical analysis method is also missing. Since it is a study paper, you should carry statistical data analysis from existing research.

There is not a comprehensive discussion section comparing the results and research literature.

Literature review needs to be up to date of current convergence technologies and mention the limitations of existing convergence techniques. 

Author Response

Dear reviewer,

Thank you for your comments, Point  by point  answers for your comments are below:

C1.)Introduction section doesn’t cover the basics of CT and applications. How the readers will understand about this study?

A1.)Basics and other applications of CT are not within the scope of the present study. This was written from the perspective of histopathology.  It was questioned  whether 3D micro-computed tomography (micro-CT) could replace the traditional histopathological examination method to provide the details regarding the tumor that will be  used by oncologists to treat  pulmonary carcinoma. However, we added the below sentence to give some perspective on the various fields that micro-CT can be used (Lines   78-81  highlighted in red).

Micro-CT is generally used to obtain precise information about the internal structure of materials. It is known to be used in the industrial field and materials science such as engineering, microprocessor production, dealing with materials such as stone and metal and  and also in examination of human tissues.

C2.) What is the aim of this study? Are you delivering any outcome through this study?

A2.) The aim  of the study was added in the end of the introduction paragraph as follows (Lines 66-70):

In this article, it was questioned  whether 3D micro-computed tomography (micro-CT) could replace the traditionally used histopathological examination method to provide the details regarding the tumor that are  necessary  in the management of pulmonary carcinoma. For this purpose, the results obtained in studies conducted with micro-CT in clinical specimens from pulmonary and other human tumor types were investigated, and the possible role of this new tool in the evaluation of lung cancer specimens was examined.

C3.) Many innovative techniques introduced through CT approach. Those are not covered in this study.

A3.) Same answer  to the first comment applies.

C4.)The statistical analysis method is also missing. Since it is a study paper, you should carry statistical data analysis from existing research.

A4.) In this paper, we tried to make predictions about the possible areas of use of micro-CT in lung cancer based on  a limited number of  existing studies. We cannot  see what kind of  statistical method can be applied here.

C5.) There is  not a comprehensive discussion section comparing the results and research literature.

A4.) There is in fact  no discussion section under the title of  discussion. However results of the previous studies are summarized in the two tables and their outcomes are discussed under the title:

  1. POTENTIAL CONTRIBUTION OF THIS IMAGING MODALITY TO THE EXAMINATION OF PULMONARY CARCINOMA SPECIMENS

C6.)Literature review needs to be up to date of current convergence technologies and mention the limitations of existing convergence techniques. 

A6.)This is out of our scope.

Reviewer 2 Report

The paper presents new and promising perspectives for tissue material diagnostics. Although the impact of radiation dose on subsequent gene mutation activity is not fully understood, micro-CT is likely to become a permanent fixture in the diagnostic toolkit of pathology, especially with the involvement of artificial intelligence. However, the introduction of micro-CT is associated with a significant increase in examination costs. Will these costs be reduced in the future? I suggest including a brief paragraph on this topic in Section 4.

Author Response

Dear reviewer,  thank you for your constructive comments. Your comments and our answer is below.

The paper presents new and promising perspectives for tissue material diagnostics. Although the impact of radiation dose on subsequent gene mutation activity is not fully understood, micro-CT is likely to become a permanent fixture in the diagnostic toolkit of pathology, especially with the involvement of artificial intelligence. However, the introduction of micro-CT is associated with a significant increase in examination costs. Will these costs be reduced in the future? I suggest including a brief paragraph on this topic in Section 4.

We added the following to section 5 (previously section 4) (Lines 302-307):

Another factor that will limit the use of microCT in pathology laboratories is the cost. The micro-CT scanner systems are widely used in different areas. The micro-CT scanners are composed of three main components; the X-ray source, the detector, and the rotary stage [104]. The cost of the system is dependent on mainly the field of view and the energy of the X-ray tube, and the systems can cost between $100,000 to over $1,000,000 according to the components used [105]. The use of a medium-sized high-definition detector reduces the overall cost of the system and thus this technique can be suitable for routine use [105].

Reviewer 3 Report

Comments to the Author

This study aims to explore the implementation of micro computed tomography (CT) in the diagnosis of Lung Carcinoma.

Minor English review is required.

I have a few comments:

1.       Could the authors state clearly the purpose of the study

2.       In introduction section: ‘’ Although micro-CT is far from being a method that is routinely used in the diagnosis of human diseases, 636 articles 76 were present in a query made on PubMed® with the keywords ("micro-CT" OR "X-ray micro-tomography" OR “micro- 77 computed tomography”) AND ("lung*" OR "pulmonary") on the 2nd of September 2023’’  in case authors aim to review the literature I would use the PRISMA guidelines

3.       Introduction is quite extensive.  May the authors consider moving tables 1 & 2 in another distinct paragraph?

4.       Could the authors remove the direct questions throughout the article and use maybe short titles?

minor english review is required - the use of some words is not ideal 

Author Response

Dear reviewer thank you  for your  valuable comments. Point by point answers  to your suggestions are below:

C1.)Could the authors state clearly the purpose of the study

A1.)A new paragraph is added  to the end of introduction (Lines 66-70):

In this article, it was questioned  whether 3D micro-computed tomography (micro-CT) could replace the traditionally used histopathological examination method to provide the details regarding the tumor that are  necessary  in the management of pulmonary carcinoma. For this purpose, the results obtained in studies conducted with micro-CT in clinical specimens from pulmonary and other human tumor types were investigated, and the possible role of this new tool in the evaluation of lung cancer specimens was examined.

C2.) In introduction section: ‘’ Although micro-CT is far from being a method that is routinely used in the diagnosis of human diseases, 636 articles 76 were present in a query made on PubMed® with the keywords ("micro-CT" OR "X-ray micro-tomography" OR “micro- 77 computed tomography”) AND ("lung*" OR "pulmonary") on the 2nd of September 2023’’  in case authors aim to review the literature I would use the PRISMA guidelines

A2.) PRISMA  guidelines were implemented and mentioned in two seperate paragraphs

(Lines 90-96)

According to the PRISMA Extension for Scoping Reviews (PRISMA-ScR) [22], 27 reviews and 7 case reports/letters were removed. Non-English (n = 10) records and studies with species option marked as other than “human” in the database (n = 383) were excluded. Of the 209 reports assessed for eligibility, only three studies were performed on human lung carcinoma specimens; the remaining 206 were non-tumoral, animal studies or irrelevant. Apart from these three studies, most of the other studies represent the research results on tumoral and non-tumoral lung pathologies performed on laboratory animals, as this promising novel imaging modality may be useful in understanding various lung diseases.

Lines 105-111

1052 articles were found in a query made on PubMed® with the keywords ("micro-CT" OR "microCT" OR "micro-computed tomography") AND ("tumor" OR "carcinoma") on the 2nd of September 2023. According to PRISMA-ScR [22], 17 reviews and 8 case reports/ letters were removed. Non-English (n = 25) records and studies with species option marked as other than “human” in the database (n = 567) were excluded. Of the 435 reports assessed for eligibility, only 26 studies were performed on human non-pulmonary tumor specimens; the remaining 409 were non-tumoral, animal studies or irrerelevant. The important features of these studies, most of which were conducted on breast carcinoma, are summarized in Table 2.

C.3) Introduction is quite extensive.  May the authors consider moving tables 1 & 2 in another distinct paragraph?

A.3) Introduction is  shortened. The second part of the introduction  and the tables  were moved to a new section:

MIcro-CT as an emergIng ImagIng modalIty In human tumor pathology

C4.) Could the authors remove the direct questions throughout the article and use maybe short titles?

A4.) All but one of the direct questions are removed  and short titles are used as suggested:

Examination steps where micro-CT can be used

Micro-CT cannot replace microscopy  in its current capacity

Tumor size and volume estimation by micro-CT

Detection of LN metastases using micro-CT

Possible role of micro-CT in the intraoperative examination  of lung tumors

IMPACT OF MICRO-CT ON CONVENTIONAL HISTOLOGICAL EXAMINATION

Round 2

Reviewer 1 Report

As per the comments, the article has been revised. I think the format is not perfect.